# Role of GDF15/MAPK14 Axis in Chondrocyte Senescence as a Novel Senomorphic Agent in Osteoarthritis

**DOI:** 10.3390/ijms23137043

**Published:** 2022-06-24

**Authors:** Pei-Wei Weng, Narpati Wesa Pikatan, Syahru Agung Setiawan, Vijesh Kumar Yadav, Iat-Hang Fong, Chia-Hung Hsu, Chi-Tai Yeh, Wei-Hwa Lee

**Affiliations:** 1Department of Orthopaedics, School of Medicine, College of Medicine, Taipei Medical University, Taipei 11031, Taiwan; wengpw@tmu.edu.tw; 2Department of Orthopaedics, Shuang Ho Hospital, Taipei Medical University, New Taipei City 23561, Taiwan; 3Graduate Institute of Biomedical Materials and Tissue Engineering, College of Biomedical Engineering, Taipei Medical University, Taipei 11031, Taiwan; 4Department of Medical Research & Education, Shuang Ho Hospital, Taipei Medical University, New Taipei City 235, Taiwan; narpatisesa@gmail.com (N.W.P.); setiawan.syahru@gmail.com (S.A.S.); vijeshp2@gmail.com (V.K.Y.); impossiblewasnothing@hotmail.com (I.-H.F.); ctyeh@s.tmu.edu.tw (C.-T.Y.); 5Division of Urology, Department of Surgery, Faculty of Medicine, Universitas Gadjah Mada/Dr. Sardjito Hospital, Yogyakarta 55281, Indonesia; 6International Ph.D. Program in Medicine, College of Medicine, Taipei Medical University, Taipei City 11031, Taiwan; 7Department of Emergency Medicine, Shuang-Ho Hospital, Taipei Medical University, New Taipei City 23561, Taiwan; 8Graduate Institute of Injury Prevention and Control, College of Public Health, Taipei Medical University, Taipei City 11031, Taiwan; 9Department of Medical Laboratory Science and Biotechnology, Yuanpei University of Medical Technology, Hsinchu City 30015, Taiwan; 10Department of Pathology, Shuang Ho Hospital, Taipei Medical University, New Taipei City 23561, Taiwan

**Keywords:** GDF15, MAPK14, osteoarthritis, cellular senescence, senotherapeutics

## Abstract

Osteoarthritis (OA) is most prevalent in older individuals and exerts a heavy social and economic burden. However, an effective and noninvasive approach to OA treatment is currently not available. Chondrocyte senescence has recently been proposed as a key pathogenic mechanism in the etiology of OA. Furthermore, senescent chondrocytes (SnCCs) can release various proinflammatory cytokines, proteolytic enzymes, and other substances known as the senescence-associated secretory phenotype (SASP), allowing them to connect with surrounding cells and induce senesce. Studies have shown that the pharmacological elimination of SnCCs slows the progression of OA and promotes regeneration. Growth differentiation factor 15 (GDF15), a member of the tumor growth factor (TGF) superfamily, has recently been identified as a possible aging biomarker and has been linked to a variety of clinical conditions, including coronary artery disease, diabetes, and multiple cancer types. Thus, we obtained data from a publicly available single-cell sequencing RNA database and observed that GDF15, a critical protein in cellular senescence, is highly expressed in early OA. In addition, GDF15 is implicated in the senescence and modulation of MAPK14 in OA. Tissue and synovial fluid samples obtained from OA patients showed overexpression of GDF15. Next, we treated C20A4 cell lines with interleukin (IL)-1β with or without shGDF15 then removed the conditioned medium, and cultured C20A4 and HUVEC cell lines with the aforementioned media. We observed that C20A4 cells treated with IL-1β exhibited increased GDF15 secretion and that chondrocytes cultured with media derived from IL-1β–treated C20A4 exhibited senescence. HUVEC cell migration and tube formation were enhanced after culturing with IL-1β-treated chondrocyte media; however, decreased HUVEC cell migration and tube formation were noted in HUVEC cells cultured with GDF15-loss media. We tested the potential of inhibiting GDF15 by using a GDF15 neutralizing antibody, GDF15-nAb. GDF15-nAb exerted a similar effect, resulting in the molecular silencing of GDF15 in vivo and in vitro. Our results reveal that GDF15 is a driver of SnCCs and can contribute to OA progression by inducing angiogenesis.

## 1. Introduction

Osteoarthritis (OA) is a common chronic bone disease characterized by deterioration and inflammation. The primary symptoms of OA include pain and mobility problems. Various variables, including age and trauma, influence the development of OA [1]. Chondrocytes are the only cells found in the cartilage tissue and play a crucial role in maintaining cartilage stability. Chondrocytes undergo various alterations throughout the onset and progression of OA, including changes in their proliferation, viability, and secretion [2]. OA is considered a multifaceted illness affecting the entire joint rather than only the cartilage or synovium. This opens new avenues for identifying and developing novel therapies as well as reprofiling candidate medications. Advanced knowledge regarding the pathogenic mechanisms of OA has elucidated the crucial functions of various new pathways that can be targeted. Because OA is a complex disease, a single treatment targeting a specific joint tissue may not be beneficial, and a single multipurpose therapy has yet to be developed [1,3].

Detailed knowledge of basic mechanical modifications is required to examine temporal fluctuations in disease progression, such as the transition from high bone resorption in the initial stages to decreased bone resorption in the end stages, and to detect changes in the pain intensity. The selection of suitable medication during specific time points of the illness may assist in the future development of personalized medicine for every patient. In addition, OA may exhibit conflicting endotypes, such as inflammatory pain, which may benefit from the combination of drugs targeting both pain and inflammation [3,4]. OA is not a distinct illness with a single pathological mechanism but involves a combination of processes and risk factors that lead to the mechanical problem of the joint. Thus, recognizing initial OA stages can be valuable for the development of efficient personalized therapies. The identification of reliable biomarkers and the development of effective imaging tools and interdisciplinary therapy regimens are critical. Cellular senescence, epigenetic changes, mitochondrial failure, dysregulated nutrition sensing, stem cell depletion, proteostasis loss, and telomere attrition have been proposed as foundational events in aging [5]. These processes might be interconnected, and some crosstalk might exist among them. Researchers have focused increasing attention on cellular senescence because the elucidation of aging-related processes underlying the development of numerous illnesses can facilitate the development of therapies.

GDF15, a member of the tumor growth factor (TGF)-β superfamily, is involved in various pathological illnesses, including coronary artery disease, diabetes, and several cancer types [6,7,8]. GDF15 was recently described as a potential biomarker of aging [9]. Furthermore, higher GDF15 levels were observed in older adults. Moreover, individuals with higher GDF15 levels had higher aspartate aminotransferase and total cholesterol (TC)/high-density lipoprotein-cholesterol (HDL-C) levels. These biological features are typically linked with aging-related diseases, such as atherosclerosis [10]. The aforementioned findings indicate that GDF15 may play a crucial role in driving chondrocyte senescence and the angiogenic microenvironment in OA.

In this study, we combined the bioinformatics approach with clinical, in vitro, and in vivo assays to understand the role of GDF15 in driving senescence in chondrocytes and its effect on angiogenesis. The results of this study revealed that the targeting of GDF15 in OA may constitute a therapeutic strategy.

## 2. Results

### 2.1. GDF15 Secretory Factor Is a Potential Driver of Chondrocyte Senescence and OA

To identify the potential driver of senescence in OA, we surveyed the publicly available scRNA-seq GEO data set GSE176308. We classified the cluster of cells into Early OA nonpainful, Early_OA painful, and Endstage_OA painful (Figure 1A, left panel). Furthermore, we examined the type of cells that were dominant in those clusters (Figure 1A, top panel). The results revealed that GDF15 was among the most highly expressed genes in the Early_OA painful cluster (Figure 1B). Our analysis indicated that GDF15 was strongly expressed in endothelial cells and chondrocytes (Figure 1C). VCAM1, IL-6, and GDF15 exhibited a similar pattern of expression (Figure 1D). Combining the aforementioned data with data obtained from the Senequest database (https://senequest.net; accessed on 8 November 2021), we determined that GDF15 drives senescence in OA (Figure 1E). To explore the pathway network in which GDF15 plays a role, we utilized the online platform OATargets (http://skeletalvis.ncl.ac.uk/OATargets/#; accessed on 8 November 2021) (Figure 1F). The results revealed that the role of GDF15 in OA remains unclear and that GDF15 may drive senescence in chondrocytes and endothelial cells through MAPK14. These preliminary data suggest that GDF15 plays a crucial role in OA progression by driving cellular senescence.

### 2.2. GDF15 Secretory Factor Is Strongly Expressed in Patients with OA

To validate our bioinformatics findings, we collected clinical samples from 12 patients with OA who underwent needle biopsy in Shuang-Ho hospital (Figure 2A). Baseline clinical characteristics of patients with osteoarthritis and control are described in Table 1.

The immunohistochemistry results revealed that GDF15 was highly expressed in the synovial membrane (SM) of the patients with OA compared with the SM of healthy individuals (Figure 2B). In addition, ELISA showed that GDF15 was strongly detected in the synovial fluid of the patients with OA compared to the synovial fluid of healthy individuals (Figure 2C). Because our bioinformatics findings suggested a possible relationship between GDF15, MAPK14 pathway, and angiogenesis, we stained SM tissues with pMAPK14, p16, and CD31 antibodies. Interestingly, we found that GDF-15, pMAPK14, p16, and CD31 are more strongly stained in OA SM tissues compared to normal SM tissues (Figure 2D). In our OA samples, we observed a positive correlation between GDF15 and pMAPK14 expression. Similarly, we noted that GDF15 was positively correlated with the senescence marker p16 and the angiogenesis marker CD31 (Figure 2E). We calculated the OARSI score of the patients and determined that the OARSI score was positively correlated with the GDF15 expression level in the synovial fluid of patients with OA (Figure 2F). These findings demonstrated the role of GDF15 in OA pathogenesis.

### 2.3. GDF15 Induced the SASP and MAPK14 Activity of Chondrocytes in a Paracrine/Autocrine Manner

We established an experimental model with chondrocyte IL-1β stimulation (10 ng/mL) and GDF15 loss of function. The C20A4 cells were divided into three treatment groups: control (shScramble), inflammation (shScramble + IL-1β), and GDF15 loss (shGDF15 + IL-1β) groups. After 48 h of incubation, the conditioned culture media of the three groups were collected and used in another batch of the C20A4 culture to examine senescence and inflammation. After 24 h of incubation, the CM of the senescent C20A4 cells (control CM, IL-1β CM, and shGDF15 CM) was used to incubate a batch of HUVEC cells. We examined the migration and tube-formation ability of cultured HUVEC cells. By using this model, we simulated the paracrine/autocrine circuit effect of GDF15 on chondrocytes and endothelial cells (Figure 3A). In this paracrine/autocrine model, the secretion of GDF15 was significantly increased in the inflammation group as compared to the GDF15-loss group (Figure 3B). The differential protein expression of GDF15 and pMAPK14 was noted in the lysate of C20A4 cells treated with the CM of the inflammation and GDF15-loss group (Figure 3C). qRT-PCR analysis of the SASP-associated markers IL-6, IL-8, MMP-13, and Cdkn1α mRNA expression showed that expression was weaker in the C20A4 cells treated with the CM of the GDF15-loss group than in the C20A4 cells treated with the CM of the inflammation group (Figure 3D). Thus, the findings of this model indicated that GDF15 may affect MAPK14 activity and SASP in chondrocytes.

### 2.4. GDF15 Drives Chondrocyte Senescence and Angiogenesis

To investigate the effect of GDF15 on chondrocyte senescence, we cultured C20A4 cells with the CM of the control, inflammation, and GDF15-loss groups for 48 h. First, we determined the protein expressions of BAX, BCL2, p16, and p21 in the C20A4 cells incubated with the CM of the inflammation and GDF15-loss groups. Senescent cells exhibit some pro-survival characteristics. Our results revealed lower BAX protein expression and higher BCL2 protein expression in the chondrocytes incubated with the CM of the inflammation group than in those incubated with the CM of the GDF15-loss group. The markers of senescence, namely, p16 and p21, were significantly overexpressed in the inflammation group compared with the GDF15-loss group (Figure 4A). Subsequently, we used the senescence marker SA-β-Gal to stain the C20A4 cells cultured with the CM of the control, inflammation, and GDF15-loss groups, respectively. The results demonstrated that the C20A4 cells cultured with the CM of the inflammation group exhibited stronger SA-β-Gal staining than did the C20A4 cells cultured with the CM of the control group (Figure 4B). Furthermore, the C20A4 cells cultured with the CM containing a lower GDF15 level exhibited weaker SA-β-Gal staining. The chondrocytes incubated with the CM containing GDF15 exhibited an increased expression of SASP. We examined the effect of chondrocyte senescence on endothelial cells by culturing HUVEC cells with the CM of the aforementioned affected chondrocytes (control CM, IL-1β CM, and shGDF15 CM) (Figure 4C). In addition, the qRT-PCR results indicated the upregulation of angiogenesis-related markers, namely VEGF, VEGFR2, and VEGFR3, in the HUVEC cells. The HUVEC cells cultured with the CM of the GDF15-loss group, which exhibited a lower senescent phenotype, exhibited lower migration and tube-formation abilities (Figure 4D). These findings indicated that GDF15 induces cell senescence and angiogenesis in OA.

### 2.5. GDF15-Targeting Monoclonal Antibody May Act as a Senomorphic Agent in OA

We examined the function of the GDF15/MAPK14 signaling pathway in OA by using the autocrine/paracrine model. We utilized GDF15-nAb, a GDF15 neutralizing antibody, to investigate the therapeutic efficacy of targeting this pathway in OA. The senescence-associated secretory phenotype (SASP) of chondrocytes plays a crucial role in the pathogenesis of OA. Thus, we first determined whether GDF15-nAb can reduce secretory phenotype of the chondrocytes incubated with the CM of the inflammation group through IL-1β induction. The results revealed that GDF15-nAb effectively inhibited secretory phenotype of chondrocyte as evidence by significant decreased of the SASP markers expression, such as IL-6, IL-8, MMP-13, and CDKN1α (Figure 5A). Moreover, the expression of SA-β-Gal as the cellular senescence marker were also inhibited following co-treatment with GDF15-nAb (Figure 5B). In addition, we investigated the effect of GDF15-nAb on DNA damage-induced cellular senescence. As expected, GDF15-nAb repressed DNA damage response following IL-1β- induced senescence as demonstrated by reduction of γH2AX, p16, and p21 (Figure 5C). These findings support the translational possibility of the GDF15 targeting strategy as senotheraputics agent in OA.

### 2.6. GDF15 Interacts with the ErbB2 Receptor to Exert Its Effects in OA

The detailed mechanism of GDF15 action in chondrocytes remains poorly understood. GDF15 interacts with the GFRAL receptor. However, the expression of this receptor in chondrocytes remains unclear. The ErbB2 receptor is expressed in chondrocytes, especially in immature chondrocytes. Thus, we utilized recombinant human GDF15 (rhGDF15) and shErbB2 to test this hypothesis. We determined that C20A4 cells expressed the ErbB2 receptor. The cells treated with rhGDF15 demonstrated increased phosphorylation of ErbB2, and this, in turn, increased the phosphorylation of MAPK14 (Figure 6A,B). Furthermore, the results of the immunoprecipitation assay revealed that GDF15 was bound to p-ErbB2 in C20A4 cells (Figure 6B). Thus, these experiments indicated that GDF15 partially exerted its effect through the ErbB2 receptor in chondrocytes.

### 2.7. GDF15-nAb Treatment Reduced Angiogenesis in the Synovial Tissue of Rat

We performed animal experiments to determine the effectiveness of GDF15-nAb. We administered an intraarticular injection of GDF15-nAb in the respective mouse groups and examined the synovial tissues of the knee joint after 28 days (Figure 7A). Compared with the control group, the GDF15-nAb-treated group exhibited decreased synovial blood vessels, synovial cell proliferation, and synovitis score and an elevated degree of synovitis (Figure 7B). We observed that GDF15-nAb significantly reduced the OARSI score in the GDF15-nAb-treated group compared with the control group (Figure 7C). Immunohistochemical staining revealed that the number of CD31^+^ cells was higher in the synovial tissues of OA rat than in those of control rat due to enhanced vascular endothelial cell proliferation. The OA group had considerably more CD31^+^ cells in the synovial tissues than did the control group; however, treatment with GDF15-nAb yielded opposite outcomes (Figure 7D,E). Similar to the effects observed in vitro, GDF15-nAb treatment markedly reduced the presence of senescent cells induced by OA in the articular cartilage. The clearance of senescent chondrocytes (SnCs) was verified by (a) a lower number of cells lacking nuclear HMGB1 and an elevated number of non-SnC-positive cells for nuclear HMGB1; (b) an elevated number of non-SnCs expressing Ki-67 and proliferating cell nuclear antigen; (c) a decrease in the number of p16^INK4a^- and MMP13-positive chondrocytes. In addition, we observed a decline in the number of Ki-67-positive cells in the synovium of OA rat; this finding is consistent with the presence of synovial SnCs, which were abolished by GDF15-nAb treatment.

## 3. Discussion

Osteoarthritis is a type of arthritis that affects the joints. Significant shifts in the metabolism, functionality, and anatomical structure of varying joints and periarticular tissues clearly define this condition. The damage affects structures including cartilage, meniscus, synovial, and subchondral bone. The most typical symptoms of osteoarthritis are mechanical pain, joint abnormalities and swelling, limitation of range of movement, joint stiffness, and motion cracks. Flaring phases with dominant inflammation, which are commonly characterized by joint swelling, a fast escalating pain, discomfort at rest, and increased stiffness, might impede the natural development of OA [2,11,12]. According this study, we demonstrated the role of GDF15 in OA. By employing a bioinformatics approach, we observed that GDF15 is among the most upregulated genes in early OA and that its expression may have induced cellular senescence in chondrocytes affected by OA through MAPK14 activation. We determined that GDF15 is highly secreted in the inflammatory condition and that the utilization of GDF15-rich media induced the expression of SASP in chondrocytes. This secretory phenotype, in turn, resulted in the migration of and tube formation in HUVEC cells, implicating the role of GDF15 in angiogenesis. Because ErbB2 was reported to be expressed in chondrocytes and GDF15 was observed to interact with this receptor in ovarian cancer, we demonstrated that this interaction was conserved in chondrocytes and may have partially explained the mechanism of the GDF15/MAPK14 signaling pathway. Translationally, we noted that GDF15-nAb, a GDF15 neutralizing antibody, effectively hindered the expression of SASP in chondrocytes in vitro and in vivo. From our own perspective, this is a pioneering study highlighting that GDF15 accelerates chondrocyte senescence and might be used as a therapeutic target for OA.

In the early and late phases of OA, inflammation targets SM, causing synovitis. The spread of inflammation in the early stage is restricted to regions near chondropathy sites and is linked with the acceleration of cartilage degeneration (chondrolysis). This finding implies that inflammation is caused by cartilage disintegration. Synovitis occurs throughout the SM in advanced OA, causing fibrosis and villi hypertrophy. Tensile load damages cartilage directly or activates chondrocytes, exacerbating aberrant concentrations of matrix metalloproteinases (MMPs) and causing reactive oxygen species (ROS) to be generated, resulting in cartilage breakdown and the efflux of microcrystals, osteochondral fragments, and extracellular matrix decomposition products in the joint space. Inflamed synovial cells that encompass synoviocytes, macrophages, and lymphocytes secrete cytokines, chemokines, lipid mediators, reactive oxygen species (ROS), and MMPs, which can directly cause deterioration of cartilage matrix components or dysregulate chondrocyte metabolic activity, culminating in a mismatch between cartilage matrix breakdown and production. Cartilaginous degradation products, as well as proinflammatory cytokines produced by chondrocytes and other joint cells, aggravate SM inflammation, creating a continuous devastating loop [13].

Angiogenesis, or the development of new capillaries from existing blood vessels, has been related to inflammation. The effector cells of the inflammation phase release proangiogenic molecules that facilitate the formation and invasion of new blood vessels, allowing inflammatory cells to invade. Proinflammatory cytokines can directly cause neovascularization or stimulate the synthesis of VEGF. The cytokines, including tumor necrosis factor (TNF-α), IL-1, IL-6, IL-15, IL-17, IL-18, oncostatin M, granulocyte colony-stimulating factor, granulocyte–macrophage colony-stimulating factor, and macrophage migration inhibitory factor play a role in angiogenesis and OA synovitis [14,15]. Angiogenesis appears to be a major factor in the persistence of OA because it plays a key role in synovial inflammation and cartilage degradation. Angiogenesis promotes the infiltration of inflammatory cells and the localization of pain receptors. Thus, inhibiting angiogenesis can control inflammation and pain in OA.

Senescence causes metabolic changes in cells, thus promoting the progression of OA over time. Senescent fibroblasts transplanted into mouse knee joints induced cartilage degradation and osteophyte development and reduced mobility, implying that senescent cells modify the synovial milieu and cause OA-like arthropathy [16]. Senescent joint cells share characteristics such as the erosion of telomeres, the enhanced expressions of p53 and the CDK inhibitors p21 and p16INK4a (p16), the enhanced formation of ROS owing to mitochondrial failure, and the enhanced expression of senescence-associated heterochromatin [17]. SASP is found in osteocytes, chondrocytes, and synovial fibroblasts. SASP is characterized by the release of proinflammatory cytokines such as IL-1, IL-17, IL-6, TNF-α, and oncostatin M, and various SASP components trigger OA-related alterations such as inflammation, bone development, and extracellular matrix destruction. Thus, discovering the phenotypic ramifications of SASP factors in joint tissues can elucidate the etiology of OA [18,19,20,21].

GDF15, like other proteins, is regulated at several levels in cells, including transcription, translation, and even translocation. From the cytoplasmic compartment, GDF15 originates as a precursor form as proGDF15 dimer before being processed and secreted as mature form of dimeric GDF15. In addition, the propeptide of GDF15, which is a cleavage product and an unprocessed proGDF15 dimer, can attach to the extracellular matrix and operate as a deposit site. Only mature GDF15 levels in the blood can be easily identified, and they are very low in healthy people; nevertheless, they are significantly raised in cancer, cardiovascular disease, liver and kidney disease, and tissue damage. Additionally, during pregnancy, GDF15 levels in the blood are significantly elevated, while GDF15 levels in the placenta are substantially elevated. Age, smoking, stress, and environmental variables are all risk factors that might elevate GDF15 concentrations. As a result, GDF15 is indicated as a biomarker for a variety of disorders and is thought to be a predictor of all-cause death [8,9,10].

GDF15, together with osteopontin and IL-8, is released by senescent endothelial cells and triggers the ROS accumulation mediated by p16 signaling [22]. Similar results were reported in senescent adult blood endothelial colony-forming cells that generated more GDF15. This phenomenon shows an advantageous paracrine action on nonsenescent endothelial cells; GDF15 binds to the activin receptor-like kinase 1 receptor and stimulates the Smad1 pathway. This would further trigger senescence across the airway epithelial cells exposed to pollutants, such as cigarette smoke [23]. The researchers used GDF15 as a mitokine to investigate how it controls age-related inflammatory responses and immunosenescence as elements of an adaptive strategy to ageing. Mitokines are soluble molecules produced under stresses such as mitochondrial stress [24]. It is known that GDF15 is involved in the inhibition of traditional T-cell stimulation and inflammatory cytokine production during senescence [25]. Despite increased levels of GDF15 expression in senescent cells, it is unclear whether its paracrine effects affect neighboring cells or whether it affects distant organs by circulating throughout the body.

## 4. Material & Methods

### 4.1. Ethical Considerations for Research

The Joint Institutional Review Board of Taipei Medical University accorded the authorization for this research (N202201135). At the time of admission, all patients provided written informed consent before undergoing diagnostic and therapeutic procedures.Throughout this study, the Helsinki Declaration’s principles were implemented. At the timeframes and dosages indicated, cells were exposed to human recombinant GDF15 (PeproTech, Rocky Hill, NJ, USA), MAPK14 inhibitor, or mouse antihuman GDF15 monoclonal antibody (R&D Systems, Minneapolis, MN, USA). Long-course treatment with GDF15 includes centrifuging the cells at 300× *g* twice weekly and resuspending them in CM containing GDF15.

### 4.2. Single-Cell RNA Sequencing Dataset Processing

A related single-cell RNA profiling dataset by Nanus et al. that previously observed diversity of cell population, including fibroblast, chondrocyte, endothelial, and other stromal cells that contributed to different stages of osteoarthritis was further analyzed to disclose specific gene expression between those cell clusters [26]. The dataset from Nanus et al. was archived in the Gene Experiment Omnibus repository with the accession number GSE176308. After downloading individual patient’s file matrix, the Seurat package (version 4.0.6) was enabled in R (version 4.0.1) to construct Seurat objects. Filtering unique characteristics and reducing low-quality mitochondrial genome were used as part of a standard pre-processing procedure. Thereafter, the Seurat object was normalized and scaled, followed by dimensional reduction and cell cluster creation using the t-Distributed Stochastic Neighbor Embedding (tSNE) method. Each cluster’s positive and negative markers were then generated and listed. To depict the amount of expression of interest markers between each cluster, an array of plots consisting of tSNE plots, dot plots, and bar plots were shown.

### 4.3. Tissue Specimens

The samples were obtained from patients at the institution who had undergone surgery for OA (*n* = 12). The patients had a diagnosis of OA on the basis of the guidelines of the Osteoarthritis Research Society International (OARSI) [27]. The OA grades of the patients were determined using the improved Mankin pathology score [28]. Normal cartilage specimens were collected as control samples from individuals who had traumatic lower-limb amputation (*n* = 6). The clinical characteristics of patients involved in analysis including patients with OA and control (*n* total = 18) were summarized in Table 1. Patients with progressive degeneration, evident osteoporosis, rheumatoid arthritis, and neoplasia lesions were excluded from the control group. The clinical specimens were then analyzed for further tissue staining.

### 4.4. Specimen Collection

The cartilage’s underlying subchondral bone was retrieved and preserved in formaldehyde. Three days later, the samples were decalcified for 14 days in formic acid solution (30%). The samples were then cut into appropriate sizes, treated with an ethanol gradient series, and placed in paraffin before being serially sectioned (5 μm). Prior to being analyzed, a portion of specimens was air-dried for approximately 6 h in an oven at 60–70 °C. Patients’ blood samples were obtained and combined with a standard anticoagulant. The samples were maintained at room temperature for 2 h, centrifuged (3000 r/min for 10 min), and then placed in a freezing tube for storage (−80 °C).

### 4.5. Hematoxylin–Eosin Staining

The paraffin-embedded samples were dewaxed with xylene and then re-suspended with graded ethanol solution. The specimens were then cleaned with water. Following coloring with Harris’ hematoxylin and eosin for 5 min, the samples were washed with water for 5 min prior being rinsed with filtered water. The slices were decolored by immersing them in hydrochloric acid and ethanol (0.5 percent) over ten seconds before being rinsed twice with tap water and once with filtered water. After rinsing, Feosin solution was used to stain sections for 40 s, and the slides were mounted using a gradient ethanol series after dehydration.

### 4.6. Safranin O/Fast Green Staining

Following a dewaxing procedure, the tissue sections were reacted for 1 min with 0.2 percent fast green solution, followed by mixing of 30 s with 1 percent acetic acid solution and 15 min with Safranin O solution. After being rinsed with 95 percent ethanol, the samples were cleaned with xylene and fixed with neutral gum before being dehydrated with a gradient ethanol sequence.

### 4.7. Immunohistochemistry Staining

The tissue specimens were dewaxed and subsequently blocked for about 30 min with goat serum. Primary antibodies against antirabbit-GDF15 [1:2000], p-MAPK14 [1:1000], CD31 [1:1000], and p16 [1:1000] were bought from Cell Signaling Technology (Beverly, MA, USA) and incubated overnight at 4 °C on tissue specimens. Following that, the specimens were reacted for 20 min at 37 °C using the secondary antibody (goat antirabbit IgG, 1:1000 dilution). Following multiple rinsing with phosphate-buffered solution, the samples were reacted with diaminobenzidine. A yellow–brown color in the nuclei or cytosol portion of cells was regarded as positive expression of GDF15, p-MAPK14, p16, and CD31. The proportion of positively stained cells was calculated as follows: overall number of positive cells/total cell count × 100.

### 4.8. Enzyme-Linked Immunosorbent Assay

The enzyme-linked immunosorbent assay test (ELISA) was used to detect the level of secreted protein in the preserved serum. The ELISA detection kit for GDF15 was procured from Abcam (cat: ab155432, Boston, MA, USA) and executed pursuant to the manufacturer guidelines. The succeeding day, the serum was retrieved, and the specimens were washed with washing buffer. The specimens were initially incubated for approximately one hour at room temperature in a reaction well before being rinsed. About 0.1 mL of enzyme-labeled antibody was administered to each well and incubated for approximately half to an hour. Each reaction well was filled with tetramethylbenzidine substrate solution, which was then incubated at 37 °C over approximately 30 min. To terminate the reaction, each well received 0.05 mL of 2 M sulfuric acid solution. The light density of each well was then calculated using an ELISA reader under absorbance of 450/630 nm.

### 4.9. Western Blotting

To isolate the protein content of OA chondrocytes, the sample was collected in an Eppendorf tube and further sonicated three times. In each lane of the gel, we inserted 30 g of the protein isolate and elevated the voltage from 80 through 120 V. The membrane was transferred in 100 V for about 60 min and then blocked for 1 h with 5% skim milk before being incubated overnight at 4 °C with primary antibodies (p16 (1:2000)), MAPK14/pMAPK14 (1:1000), and p21 (1:1000) which was provided by Cell Signaling Technology (Beverly, MA, USA). Other primary antibodies, which including γH2AX [1:5000], ErbB2 [1:1000], and pErbB2 [1:1000], were obtained from Abcam (Boston, MA, USA). Before incubating the membranes with the secondary antibody for an hour, membranes were rinsed three times with PBST detergent for about 5 min. After incubation with secondary antibody, the membranes were washed with PBST and viewed using the chemiluminescence reagent on the Bio-Rad Gel Doc EZ Imager (Bio-rad, Hercules, CA, USA). Image J was employed for subsequent analysis. List of the primary antibody was described in Appendix A.

### 4.10. Quantitative Real-Time Polymerase Chain Reaction

Total RNA was extracted and kept at −80 °C using the Trizol reagent (Invitrogen, Carlsbad, CA, USA). The absorbance at 260/280 nm and RNA concentration were examined using a UV spectrophotometer. The primers were constructed using VEGF, VEGFR2, VEGFR3, IL-8, IL-6, MMP13, and CDKN1a sequences from Genbank (Invitrogen, Carlsbad, CA, USA). To generate reverse transcription cDNA of the RNAs, a reverse transcription kit was employed (Takara, Tokyo, Japan). The real-time polymerase chain reaction (PCR) system from TransGen Biotech was adopted. For PCR, a 20-μL volume of the master mix was applied. The samples were predenaturated for 10 min before being denatured for 30 s (40 cycles, 95 °C), annealed for 60 s (58 °C), and extended for 30 s (72 °C). The housekeeping control was glyceraldehyde phosphate dehydrogenase (GAPDH). A solubility curve was used to analyze the repeatability of PCR measurement. We calculated the target gene expression level semi-quantitatively by calculating 2^−^^△△Ct^ from the mean Ct value.

### 4.11. In Vivo OA Rat Model

Female Wistar rats which were aged 12 weeks old and weighed around 300 g were purchased from BioLASCO Taiwan Co., Ltd. and were pathogen-free. Further investigation was implemented adhering to a protocol accepted by the Taipei Medical University Laboratory Animal Committee (protocol LAC-2020-0146). In our lab, the OA model was established according to the previously evaluated technique. To prepare this model, 200 µL of 4% papain solution and 100 µL of 0.03 mol L-1 L-cysteine solution in distilled water were needed. After mixing, the reaction was incubated for 30 min. The rat’s right knee joint was injected with 25 µL of the mixture. Intra-articular doses were repeated on days 1, 4, and 7 to produce osteoarthritis. Upon successful induction of OA, the rats were randomly separated into two groups of five animals each. The further intervention was randomly assigned to receive either the vehicle (saline) for controls, or neutralizing-antibody therapy (GDF15-nAb) for the treatment group. In this study, GDF15-nAb (10 μg/mL) was injected into the joint through a 50 μL intraarticular injection (or saline alone as a control). Following 8 weeks, all rats were euthanized, and samples of knee tissue from all groups of rats were examined histologically and immunohistochemically for signs of disease.

### 4.12. Statistics

For each numerical variable, the mean and standard error of the mean were computed. The categorical variables were represented as frequency and percentage. To assess discrepancy across groups, an analysis of variance (ANOVA) was used. Statistical significance was indicated by a *p* value below 0.05. All tests were conducted in triplicate and analyzed using GraphPad Prism 8.0 (San Diego, CA, USA).

## 5. Conclusions

In summary, our study demonstrated that the development of cellular senescence in OA is driven by GDF15 (Figure 8). Cells expressing SASP secrete inflammatory factors and degradative enzymes that worsen OA by inducing angiogenesis. The removal of the senescent phenotype by using the GDF15 neutralizing antibody could be a therapeutic approach for age-related degenerative joint diseases.

## Figures and Tables

**Figure 1 ijms-23-07043-f001:**
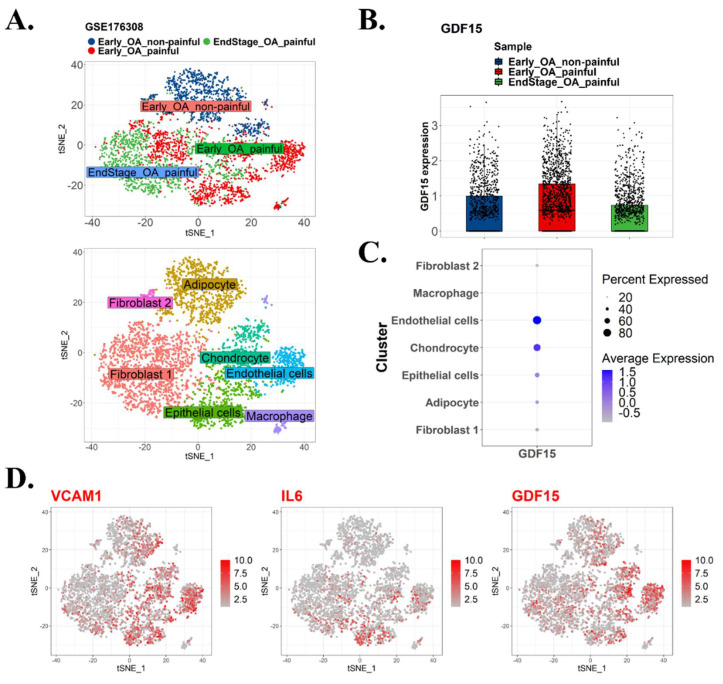
scRNA analysis depicting the overexpression of GDF15 in OA and its potential role in the MAPK14 pathway. (**A**) scRNA-seq analysis of the GEO data set GSE176308. Cells were clustered into Early_OA nonpainful, Early_OA painful, and Endstage_OA painful (left panel). Clusters of cells were identified (right panel). (**B**) GDF15 is highly expressed in the Early_OA painful cluster. (**C**) List of GDF15 expression based on cell cluster types. (**D**) Expression patterns of VCAM1, IL6, and GDF15. (**E**) Table listing overexpressed genes in Early_OA painful and their potential link to senescence based on the Senequest database (https://senequest.net; accessed on 8 November 2021). (**F**) Genes related to GDF15 in OA as predicted using the online platform OATargets (http://skeletalvis.ncl.ac.uk/OATargets/#; accessed on 8 November 2021).

**Figure 2 ijms-23-07043-f002:**
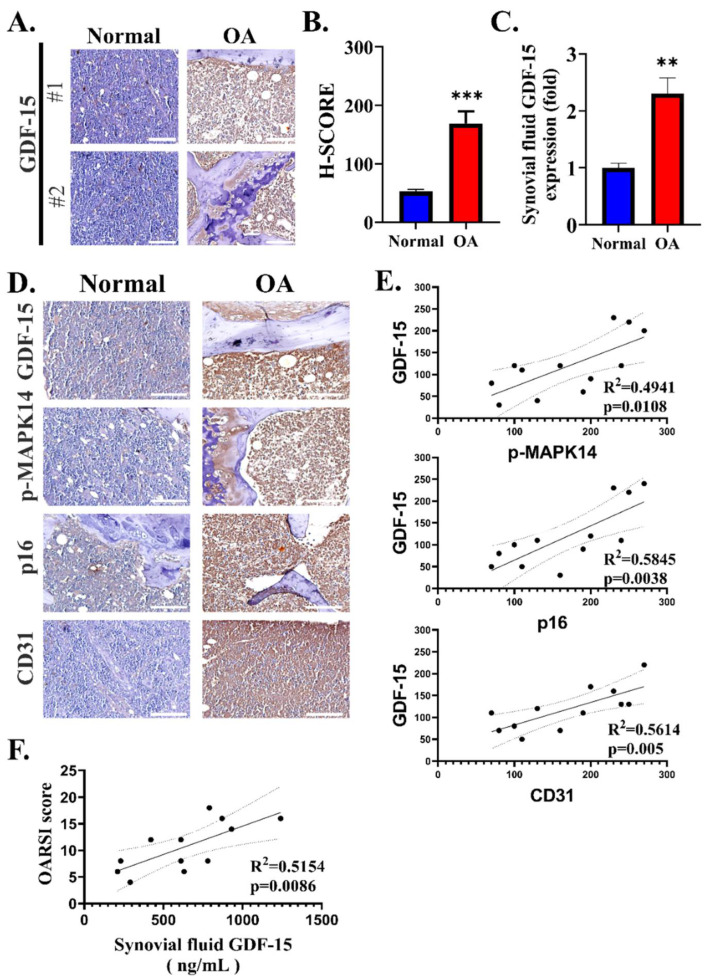
GDF15 is highly expressed in the tissue and synovial fluid of patients with OA. (**A**–**C**) Comparative study of the expression of GDF15 in healthy individuals and OA patients. (**D**–**F**) Correlation analysis between GDF15 and MAPK14, p16 as senescence marker, CD31 as angiogenesis marker, and the severity score of OA. (**A**) Representative image of the staining of GDF15 in the synovial membrane. Scale bar: 100 μm (**B**) H score quantification through GDF15 immunohistochemistry staining in the synovial membrane of healthy individuals and patients with OA. (**C**) ELISA for GDF15 detection in the synovial fluid of healthy individuals and patients with OA. (**D**) Immunohistochemistry staining comparing GDF15, pMAPK14, p16, and CD31 in the synovial membrane of healthy individuals and patients with OA. Scale bar: 100 μm (**E**) Moderate correlation of GDF15 expression with pMAPK14 (R^2^ = 0.494), p16 (R^2^ = 0.584), and CD31 (R^2^ = 0.561) in the synovial membrane tissues of healthy individuals and patients with OA. (**F**) Moderate correlation of GDF15 expression in the synovial fluid and patients’ OARSI scores (R^2^ = 0.494). ** *p* < 0.01 and *** *p* < 0.001.

**Figure 3 ijms-23-07043-f003:**
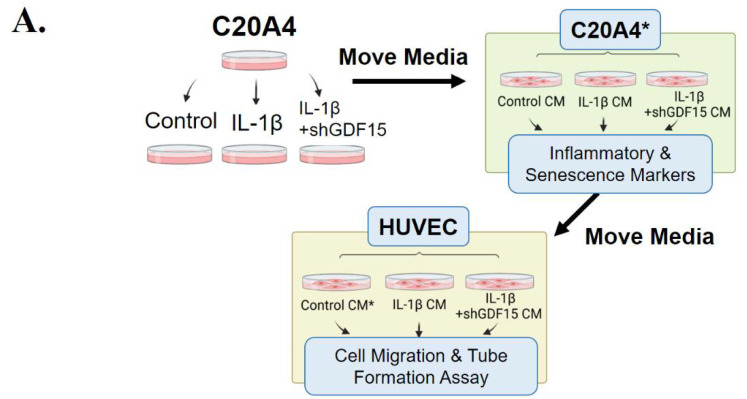
GDF15 affects chondrocytes in an autocrine/paracrine manner. (**A**) Schematic of the establishment of the GDF15 paracrine/autocrine circuit experimental model. (**B**) ELISA analysis of the GDF15 level in the medium of IL-1β-stimulated C20A4 cells transfected with shGDF15 (GDF15-loss group) compared with shScramble (Inflammation group). (**C**) Western blot analysis of C20A4 cells incubated in the conditioned medium of the inflammation group and GDF15-loss group. (**D**) qRT-PCR analysis of the SASP-associated markers IL-6, IL-8, MMP-13, and Cdkn1α mRNA expression levels in C20A4 cells incubated in the conditioned media of the inflammation and GDF15-loss groups. *** *p* < 0.001.

**Figure 4 ijms-23-07043-f004:**
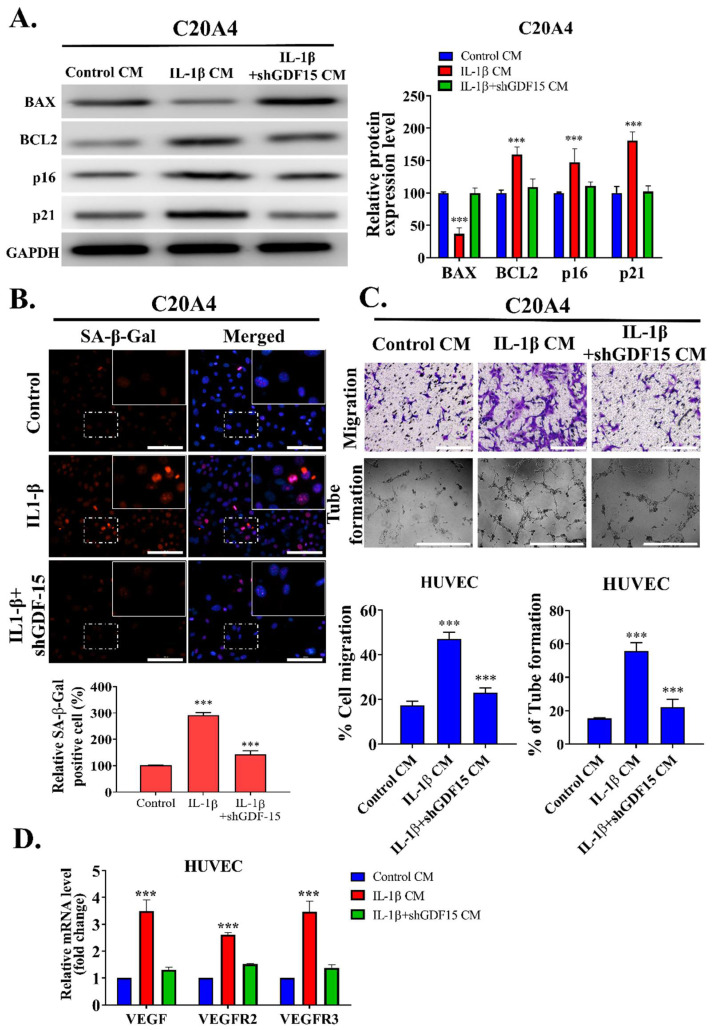
Effect of GDF15 on C20A4 transformation to senescence and SASP and its effect on endothelial cells. (**A**) Western blot protein analysis of BAX, BCL2, p16, p21 in C204 cells treated with conditioned media (Control CM, IL-1β CM, and IL-1β + shGDF15 CM). (**B**) Lower level of GDF15 resulted in weaker SA-β-Gal staining in C20A4 cells, implicating their role in chondrocyte senescence. Scale bar: 10 μm (**C**) Migration (upper panel) and tube formation (lower panel) of HUVEC cells treated with C204 conditioned media (Control CM, IL-1β CM, and IL-1β + shGDF15 CM). Scale bar: 100 μm (**D**) qRT-PCR of angiogenesis markers of HUVEC cell line incubated with conditioned media (Control CM, IL-1β CM, and IL-1β + shGDF15 CM). from the GDF15 paracrine/autocrine circuit experimental model. *** *p* < 0.001.

**Figure 5 ijms-23-07043-f005:**
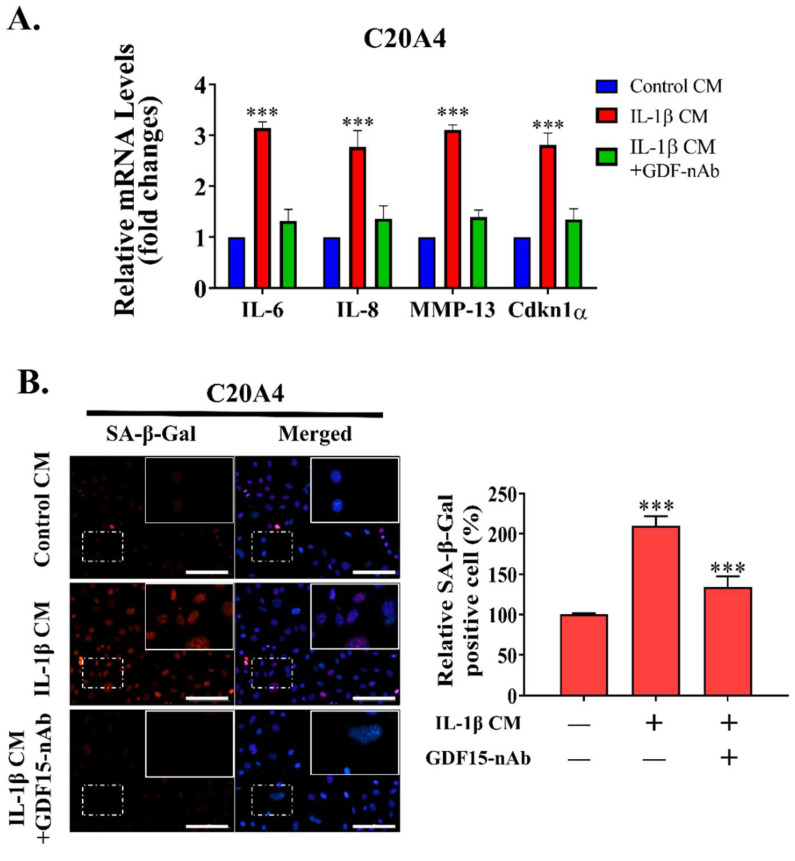
GDF15-nAb treatment attenuates chondrocyte senescence and SASP. (**A**) GDF15-nAb (1 μM/24 h) inhibited the development of the SASP markers IL-6, IL-8, MMP-13, and Cdkn1a, as observed in the qRT-PCR assay. (**B**) SA-β-GAL staining in C20A4 decreased after treatment with GDF15-nAb. Scale bar: 10 μm (**C**) Western blot assay demonstrating the effect of GDF15-nAb on chondrocytes treated with GDF15-nAb. *** *p* < 0.001.

**Figure 6 ijms-23-07043-f006:**
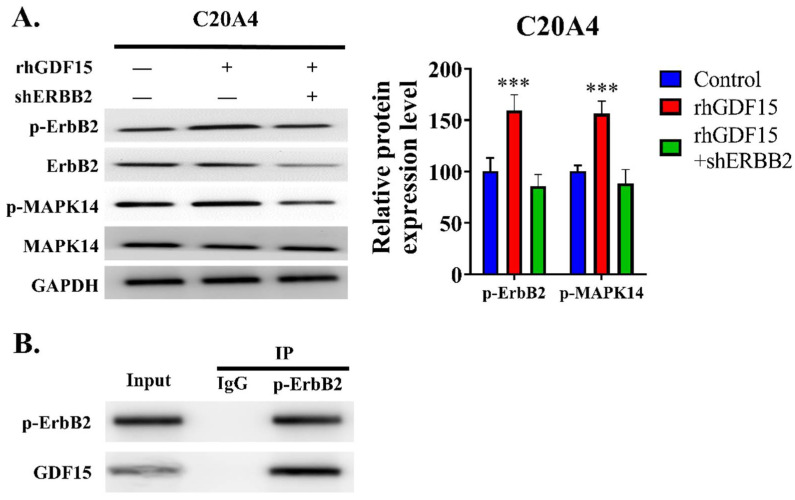
GDF15 interacts with ErbB2 in chondrocytes. (**A**) Western blot protein analysis of ErbB2, pErbB2, MAPK14, and pMAPK14 in C20A4 cells treated with or without recombinant human GDF15 (rhGDF15; 10 ng/mL) for 24 h and shERBB2. (**B**) GDF15 was immunoprecipitated from the cell lysates of IL-1β-induced C20A4, and the immunoprecipitation product was immunoblotted with an anti-p-ErbB2 antibody. *** *p* < 0.001.

**Figure 7 ijms-23-07043-f007:**
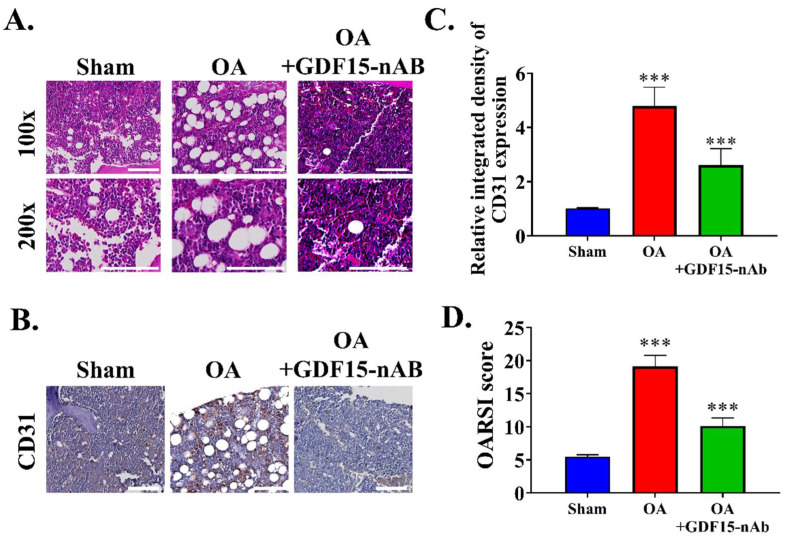
GDF15-nAb inhibits angiogenesis and attenuates posttraumatic OA in rat. (**A**) Hematoxylin and eosin staining of OA rat after GDF15-nAb therapy (scale bar, 50 or 100 μm). Scale bar: 100 μm (**B**) IHC examination of CD31 detection and positive region of knee joints from treated ACLT rat (Scale bar, 25 μm). Scale bar: 100 μm (**C**) OARSI score quantification of each groups. (**D**) Quantification of CD31-positive cells in each group. (**E**) Representative immunostaining images of HMGB1 (*n* = 5 for each group), p16INK4a (*n* = 5 for each group), Ki-67 (*n* = 5 for each group), and MMP13 (*n* = 5 for each group) in the articular cartilage from no surgery (*n* = 5), and ACLT rat treated with vehicle (*n* = 5) or GDF15-nAb (*n* = 5). Scale bar: 100 μm. *** *p* < 0.001.

**Figure 8 ijms-23-07043-f008:**
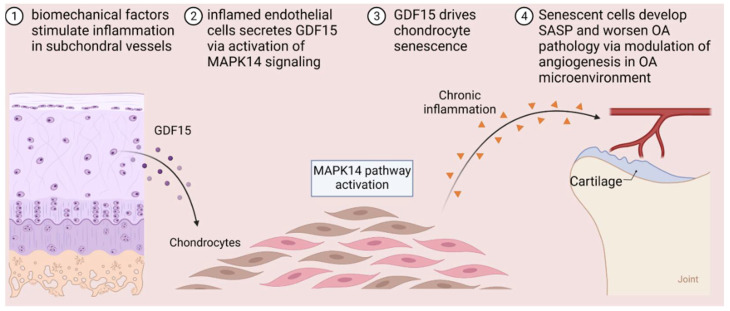
Schematic figure of cellular senescence induced by GDF15/MAPK14 signaling axis in chondrocytes of osteoarthritis.

**Table 1 ijms-23-07043-t001:** Baseline characteristics of patients with osteoarthritis (OA) and control in Shuang-Ho Hospital.

Clinical Characteristics		OA Patients (*n* = 12)	Control (*n* = 6)
Gender [n (%)]	Male	4 (33.3%)	4 (66.7%)
	Female	8 (66.7%)	2 (33.3%)
Age (years) [mean ± SD]		75.1 ± 10.3	47.6 ± 5.1
Pain severity [n (%)]	Mild	3 (25.0%)	-
	Mod/Severe	9 (75.0%)	-
OA Grade [n (%)]	1	0 (0%)	-
	2	3 (25%)	-
	3	4 (33.3%)	-
	4	5 (41.7%)	-

Pain severity: A 10-levels Numerical rating scales, categorized into ordinal groups (1–3: Mild; 4–6: Moderate; 7–10: Severe).

## Data Availability

The datasets used and analyzed in the current study are publicly accessible as indicated in the manuscript.

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
