# Peer review of "Role of GDF15/MAPK14 Axis in Chondrocyte Senescence as a Novel Senomorphic Agent in Osteoarthritis"

_ijms, 2022, doi:10.3390/ijms23137043_

Round 1

Reviewer 1 Report

The MS contains valuable results about the chondrocyte senescence and progress of osteoarthritis. The approach, results, and conclusion are sound and reliable. By the way there are some minor concerns.

1. Expressions in abstract need to be improved. Would you rephrase some sentences from line 28~41? Please don't use 'we' too much. And some are unnatural.

2. In abstract, growth differentiation factor 15 must be given at the first.

3. Check the format of references

4. Info of manufacturers are wrong. Revise them.

5. I am worrying about the visibility of words used in the figures. Would you increase the sizes?

6. You have used at least two different statistical significance indications in Fig 2. The legends must be improve to specify them.

Thank you

Author Response

We thank the reviewer for carefully reading our manuscript and providing valuable comments. We accordingly response the questions raised by the Reviewer as follows:

Point-by-point responses to reviewer’s comments:

Dear Reviewer,

Co-authors and I very much appreciated the encouraging, critical, and constructive comments on this manuscript by the reviewer. The comments have been very thorough and useful in improving the manuscript. We strongly believe that the comments and suggestions have increased the scientific value of the revised manuscript by many folds. We have taken them fully into account in revision. We are submitting the corrected manuscript with the suggestion incorporated in the manuscript. The manuscript has been revised as per the comments given by the reviewer, and our responses to all the comments are as follows:

# General Comments to Editor and all Reviewers

Please kindly find several updates in our respective draft as follows:

  1. Layout adjustment was done according to current official IJMS layout (Introduction, Result, Discussion, Methods, and Conclusion).
  2. Reference number ordering was automatically changed due to our adjustment according to IJMS layout (Introduction, Result, Discussion, Methods, Conclusion)
  3. Additional 4 references that previously were unrecognized by our citation manager (all current references in total: 40). Those additional references were mostly cited in the discussion part, such as:
  4. Henrotin, Y.; Pesesse, L.; Lambert, C. Targeting the synovial angiogenesis as a novel treatment approach to osteoarthritis. Ther Adv Musculoskelet Dis 2014, 6, 20-34, doi:10.1177/1759720X13514669.
  5. Robinson, W.H.; Lepus, C.M.; Wang, Q.; Raghu, H.; Mao, R.; Lindstrom, T.M.; Sokolove, J. Low-grade inflammation as a key mediator of the pathogenesis of osteoarthritis.
  6. Bannuru, R.R.; Osani, M.C.; Vaysbrot, E.E.; Arden, N.K.; Bennell, K.; Bierma-Zeinstra, S.M.A.; Kraus, V.B.; Lohmander, L.S.; Abbott, J.H.; Bhandari, M.; et al. OARSI guidelines for the non-surgical management of knee, hip, and polyarticular osteoarthritis. Osteoarthritis and Cartilage 2019, 27, 1578-1589, doi:10.1016/j.joca.2019.06.011.
  7. Pauli, C.; Whiteside, R.; Heras, F.L.; Nesic, D.; Koziol, J.; Grogan, S.P.; Matyas, J.; Pritzker, K.P.H.; D'Lima, D.D.; Lotz, M.K. Comparison of cartilage histopathology assessment systems on human knee joints at all stages of osteoarthritis development. Osteoarthritis and cartilage 2012, 20, 476-485, doi:10.1016/j.joca.2011.12.018.
  8. Additional methods that were previously not mentioned in the first draft, such as “4.2 Single-cell RNA Sequencing Dataset Processing”
  9. Additional keywords: GDF15; MAPK14; osteoarthritis; cellular senescence; senotherapeutics

Reviewer #1:

Comments and Suggestions for Authors

Journal: IJMS

[IJMS] Manuscript ID: ijms-1763348

Title: Role of GDF15/MAPK14 Axis in Chondrocyte Senescence as a Novel Senomorphic Agent in Osteoarthritis

Authors: Pei-Wei Weng, Vijesh Kumar Yadav, Narpati Wesa Pikatan, Iat-Hang Fong, I-Hsin Lin*, Chi-Tai Yeh and Wei-Hwa Lee *

Revision:

The MS contains valuable results about the chondrocyte senescence and progress of osteoarthritis. The approach, results, and conclusion are sound and reliable. By the way there are some minor concerns.

A: We would like to thank the Reviewer for the thorough reading of our manuscript as well as the valuable comments and appraisal. We feel that the comments and suggestions will further be helped in strengthening our manuscript.

Q1: Expressions in abstract need to be improved. Would you rephrase some sentences from line 28~41? Please don't use 'we' too much. And some are unnatural.

A1:  We thank the reviewer for highlighting this point, we have rewritten the updated abstract in the current manuscript as per the suggestions in line 24 to 47.

Abstract: Osteoarthritis (OA) is most prevalent in older individuals and exerts a heavy social and economic burden. However, an effective and noninvasive approach to OA treatment is currently not available. Chondrocyte senescence has recently been proposed a key pathogenic mechanism in the etiology of OA. Furthermore, senescent chondrocytes (SnCCs) can release various proinflammatory cytokines, proteolytic enzymes, and other substances known as the senescence-associated secretory phenotype (SASP), allowing them to connect with surrounding cells and induce senesce. Studies have shown that the pharmacological elimination of SnCCs slows the progression of OA and promotes regeneration. Growth differentiation factor 15 (GDF15), a member of the tumor growth factor (TGF) superfamily, has recently been identified as a possible aging biomarker and has been linked to a variety of clinical conditions, including coronary artery disease, diabetes, and multiple cancer types. Thus, we obtained data from a publicly available single-cell sequencing RNA database and observed that GDF15, a critical protein in cellular senescence, is highly expressed in early OA. In addition, GDF15 is implicated in the senescence and modulation of MAPK14 in OA. Tissue and synovial fluid samples obtained from OA patients showed overexpression of GDF15. Next, we treated C20A4 cell lines with interleukin (IL)-1β with or without shGDF15 then removed the con-ditioned medium, and cultured C20A4 and HUVEC cell lines with the aforementioned media. We observed that C20A4 cells treated with IL-1β exhibited increased GDF15 secretion and that chondrocytes cultured with media derived from IL-1β–treated C20A4 exhibited senescence. HUVEC cell migration and tube formation were enhanced after culturing with IL-1β-treated chondrocyte media; however, decreased HUVEC cell migration and tube formation were noted in HUVEC cells cultured with GDF15-loss media. We tested the potential of inhibiting GDF15 by using a GDF15 neutralizing antibody, GDF15-nAb. GDF15-nAb exerted a similar effect, resulting in the molecular silencing of GDF15 in vivo and in vitro. Our results reveal that GDF15 is a driver of SnCCs and can contribute to OA progression by inducing angiogenesis.

Q2: In abstract, growth differentiation factor 15 must be given at the first.

A2:  We appreciate the reviewer's valuable suggestions and thank you for bringing up this good point. In this current manuscript we have incorporated the reviewer’s suggestion and discussed GDF15 earlier in the abstract.

Corrected Sentence: … Growth differentiation factor 15 (GDF15), a member of the tumor growth factor (TGF) superfamily, has recently been identified as a possible aging biomarker and has been linked to a variety of clinical conditions, including coronary artery disease, diabetes, and multiple cancer types.

Q3: Check the format of references

A3:  We appreciate the reviewer's highlight on this issue. We have revised the reference format with the IJMS format. Please kindly refer to the revised manuscript.

Corrected references format:

  1. Quicke, J.G.; Conaghan, P.G.; Corp, N.; Peat, G. Osteoarthritis year in review 2021: epidemiology & therapy. Osteoarthritis Cartilage 2021, doi:10.1016/j.joca.2021.10.003.
  2. Young, D.A.; Barter, M.J.; Soul, J. Osteoarthritis year in review: genetics, genomics, epigenetics. Osteoarthritis Cartilage 2021, doi:10.1016/j.joca.2021.11.004.
  3. Panikkar, M.; Attia, E.; Dardak, S. Osteoarthritis: A Review of Novel Treatments and Drug Targets. Cureus 2021, 13, e20026, doi:10.7759/cureus.20026.

Q4: Info of manufacturers are wrong. Revise them.

A4:  We thank the reviewer for the thorough reading of the manuscript. We have revised the manufacturers information. Please kindly refer to the revised material & methods section for ELISA.

Corrected sentence: ELISA detection kit for GDF15 was purchased from Abcam (cat: ab155432, Boston, MA, USA) and performed according to the manufacturer’s instructions.

Q5: I am worrying about the visibility of words used in the figures. Would you increase the sizes?

A5: We appreciate the reviewer for this critical issue. We have revised the font sizes of the figures. Please kindly refer to the revised figures 1 to 8 in the manuscript.

Q6: You have used at least two different statistical significance indications in Fig 2. The legends must be improved to specify them.

A6: We appreciate the reviewer for this important issue. We have updated the figure legend to specify both different type of statistical analysis used in this figure. Please kindly refer to the revised figure legend 2 in the manuscript.

Corrected figure legend:

Figure 2. GDF15 is highly expressed in the tissue and synovial fluid of patients with OA. (A-C) Comparative study of the expression of GDF15 in healthy individuals and OA patients. (D-F) Correlation analysis between GDF15 and MAPK14, p16 as senescence marker, CD31 as angiogen-esis marker, and the severity score of OA. (A) Representative image of the staining of GDF15 in the synovial membrane. (B) H score quantification through GDF15 immunohistochemistry staining in the synovial membrane of healthy individuals and patients with OA. (C) ELISA for GDF15 detec-tion in the synovial fluid of healthy individuals and patients with OA. (D) Immunohistochemistry staining comparing GDF15, pMAPK14, p16, and CD31 in the synovial membrane of healthy indi-viduals and patients with OA. (E) Moderate correlation of GDF15 expression with pMAPK14 (R2= 0.494), p16 (R2= 0.584), and CD31 (R2= 0.561) in the synovial membrane tissues of healthy individuals and patients with OA. (F) Moderate correlation of GDF15 expression in the synovial fluid and patients’ OARSI scores (R2= 0.494).

Reviewer 2 Report

The immunocytochemical figures are of insufficient quality. A higher magnification and resolution would be appropriate. 

The legend of Fig. 4 is completely wrong. The figures are  addressed correctly within the manuscript text.

Materialand Methods:

Are the 12 patients included in the study subdivided in 3 equally large groups?

How many persons beloged to the healthy control group?

Were all 12 + normal controls analyzed immuncytochemically?

A short description of bioinformatic analysis should be given.

Typing errors: 

line   44: include and mobility problems

line 112:  F  eosin

line 264:   scheme of or schematic diagram  but not schematic of

line 346:   rat groups not mouse!

line 375:   The first word is missing (osteoarthritis)

Author Response

Reviewer #2:

Q1: The immunocytochemical figures are of insufficient quality. A higher magnification and resolution would be appropriate. 

A1: We thank the reviewer for highlighting this issue. We have changed the immunocytochemical figures with ones with better image resolution. Please kindly refer to the revised figure 2A, 4B & 5B.

Q2: The legend of Fig. 4 is completely wrong. The figures are addressed correctly within the manuscript text.

A2:  We appreciate the reviewer's valuable comment. We have revised the legend as per your suggestion. Please kindly refer to figure legend 4.

Corrected Figure Legend:

Figure 4. Effect of GDF15 on C20A4 transformation to senescence and SASP and its effect on endothelial cells. (A) Western blot protein analysis of BAX, BCL2, p16, p21 in C204 cells treated with conditioned media (Control CM, IL-1β CM, and IL-1β+shGDF15 CM). (B) Lower level of GDF15 resulted in weaker SA-β-Gal staining in C20A4 cells, implicating their role in chondrocyte senescence. (C) Migration (upper panel) and tube formation (lower panel) of HUVEC cells treated with C204 conditioned media (Control CM, IL-1β CM, and IL-1β+shGDF15 CM). (D) qRT-PCR of angiogenesis markers of HUVEC cell line incubated with conditioned media (Control CM, IL-1β CM, and IL-1β+shGDF15 CM). from the GDF15 paracrine/autocrine circuit experimental model. 

Q3-6:

Are the 12 patients included in the study subdivided in 3 equally large groups?

How many persons belonged to the healthy control group?

Were all 12 + normal controls analyzed immuncytochemically?

A: We thank the reviewer for this very critical comment. Because all those three questions refer to same section, we would like to answer those altogether. We are sorry for any confusion regarding this issue. In the current study we recruited a total of 24 patients that consists of 12 Osteoarthritis patients and 12 healthy patients. All the clinical specimen obtained from OA and control were subsequently performed for further tissue staining which the results were provided in figure 2. We have added a table detailing the patients’ characteristics in the revised manuscript. Please kindly refer to the newly provided Table1.

Updated materials & methods:

4.3. Tissue specimens

Samples were obtained from patients with OA who underwent surgery at the institution (n=12). The patients had a diagnosis of OA on the basis of the guidelines of the Osteoarthritis Research Society International (OARSI) [14]. The OA grades of the patients were determined using the improved Mankin pathology score [15]. Normal cartilage specimens were collected as control samples from individuals who had traumatic lower-limb amputation (n=6). The clinical characteristics of patients involved in analysis including patient with OA and control (n total = 18) were summarized in Table 1. Patients with progressive degeneration, evident osteoporosis, rheumatoid arthritis, and neoplasia lesions were excluded from the control group. The clinical specimens were then analyzed for further tissue staining.

New Table 1:

Table 1. Baseline characteristics of patients with osteoarthritis (OA) and control in Shuang-Ho Hospital

Clinical Characteristics

OA Patients (n=12)

Control (n=6)

Gender [n (%)]

Male

4 (33.3%)

4 (66.7%)

Female

8 (66.7%)

2 (33.3%)

Age (years) [mean ± SD]

75.1 ± 10.3

47.6 ± 5.1

Pain severity [n (%)]

Mild

3 (25.0%)

-

Mod/Severe

9 (75.0%)

-

OA Grade [n (%)]

1

0 (0%)

-

2

3 (25%)

-

3

4 (33.3%)

-

4

5 (41.7%)

-

Pain severity: A 10-levels Numerical rating scales, categorized into ordinal groups (1–3: Mild; 4–6: Moderate; 7–10: Severe)

Q7: A short description of bioinformatic analysis should be given.

A7: We thank the reviewer for the astute observation. We have followed the reviewer’s suggestion and added section “4.2 Single-cell RNA Sequencing Dataset Processing” in the material & methods section. Please kindly refer to the revised materials and methods.

Updated materials & methods:

4.2 Single-cell RNA Sequencing Dataset Processing

Related single-cell RNA profiling dataset by Nanus et al. that previously observed diversity of cell population, including fibroblast, chondrocyte, endothel, and other stromal cells that contributed to different stages of osteoarthritis was further analyzed to disclose specific gene expression between those cell clusters.[24] The dataset from Nanus et al. was archived in the Gene Experiment Omnibus repository with the accession number GSE176308. After downloading individual patient's file matrix, the Seurat package (version 4.0.6) was enabled in R (version 4.0.1) to construct Seurat objects. Filtering unique characteristics and reducing low-quality mitochondrial genome were used as part of a standard pre-processing procedure. Thereafter, the Seurat object was normalized and scaled, followed by dimensional reduction and cell cluster creation using the t-Distributed Stochastic Neighbor Embedding (tSNE) method. Each cluster's positive and negative markers were then generated and listed. To depict the amount of expression of interest markers between each cluster, an array of plots consisting of tSNE plots, dot plots, and bar plots were shown.
